# The Importance of Topographical Recognition of Pulmonary Arteries in Diagnostics and Treatment of CTEPH, Based on an Analysis of a Dissected Case Model—A Pilot Study

**DOI:** 10.3390/diagnostics14151684

**Published:** 2024-08-03

**Authors:** Matiss Zicans, Dzintra Kazoka, Mara Pilmane, Andris Skride

**Affiliations:** 1Faculty of Medicine, Rīga Stradiņš University, LV-1007 Riga, Latvia; 2Institute of Anatomy and Anthropology, Rīga Stradiņš University, LV-1010 Riga, Latvia; dzintra.kazoka@rsu.lv (D.K.); mara.pilmane@rsu.lv (M.P.); 3Department of Internal Diseases, Rīga Stradiņš University, LV-1007 Riga, Latvia; andris.skride@rsu.lv; 4Pauls Stradiņš Clinical University Hospital, LV-1002 Riga, Latvia

**Keywords:** pulmonary arteries, anatomy, dissection, chronic thromboembolic pulmonary hypertension, angiogram

## Abstract

Background: Knowledge of the anatomy of pulmonary arteries is essential in many invasive procedures concerning pulmonary circulation. In the diagnosis and treatment of chronic thromboembolic pulmonary hypertension (CTEPH), two-dimensional (2D) pulmonary angiography is used. Recognizing the topographic course of the pulmonary arteries and understanding the status in three dimensions (3D) is paramount. This study aimed to evaluate and describe the branching variant of pulmonary arteries in a single case, as well as morphological parameters of the segmental arteries, like length, diameter and branching angles. Methods: One pair of embalmed human cadaver lungs was dissected by a scalpel and surgical forceps and was measured up to the subsegmental arteries. Results: The diameters (ranging from 3.04 to 9.29 mm) and lengths (ranging from 9.09 to 53.91 mm) of the pulmonary segmental arteries varied. The proximal branching angles were wide and close to perpendicular, while distally, the angles between the segmental and subsegmental arteries were narrower (30–45°). Upon evaluating the branching, rare variations were identified and delineated, notably in the lower lobes of both lungs. Conclusions: Utilizing knowledge and data in clinical settings is instrumental for effectively diagnosing and treating CTEPH. Further research is required to explore the complications in invasive procedures related to various anatomical variations.

## 1. Introduction

The pulmonary circulatory system constitutes an intricate network of blood vessels that is detailed and structured to facilitate the efficient exchange of gases, thereby enabling the delivery of oxygen to the organs and tissues of the human body [1]. Additionally, due to the numerous variations in the arteries, physicians must have a comprehensive knowledge of the anatomy to minimize the risk of adverse intraoperative procedures [2]. Clinicians, surgeons and radiologists must thoroughly understand the normal anatomy of the pulmonary vessels. The branching and typical imaging appearance of the pulmonary arterial tree are crucial for diagnosing and characterizing various pathologies of the pulmonary arteries. The complex variations of blood vessels within the traditional anatomical borders present significant challenges for medical specialists, particularly those specializing in cardiothoracic surgery [3].

The pulmonary trunk originates from the heart’s right ventricle and initially moves forward, then to the back and side on the left of the ascending aorta. At the level of the aortic arch, the pulmonary trunk divides into the right and left pulmonary arteries. The right pulmonary artery passes under the aortic arch and enters the hilum of the right lung in front of the right bronchus. Meanwhile, the left pulmonary artery enters the hilum of the left lung above the left bronchus.

The branching patterns of these blood vessels differ between the lungs due to their anatomical variations, with the most significant variability reported in the branching of segmental and subsegmental arteries [4]. Determination of the branches is an essential aspect when attempting to assess the pulmonary vascular tree. Studying the distribution of flow resistance in the arterial tree of the lung requires a comprehensive analysis of the various quantitative aspects, including the exact numbers and lengths of each branch. This task is particularly challenging due to the irregularity of the arterial tree’s structure. Identifying systematic distributed structures introduces complexity that may pose challenges for conventional morphometric research methods.

Various disorders can affect the pulmonary arteries, occasionally incidentally discovered during imaging for other reasons or as part of the work-up for symptomatic patients. Chronic thromboembolic pulmonary hypertension (CTEPH) experiences morphologic changes in the vasculature [5]. It is a rare disease with an annual incidence of 3.1–6.0 and a prevalence of 25.8–38.4 cases per million [6]. It is diagnosed via invasive pulmonary angiography (PA) combined with right heart catheterization. CTEPH is also one of the pulmonary hypertension (PH) subtypes, representing Group 4 (based on the World Health Organization’s (WHO) classification) which can potentially be cured [7]. Other groups include pulmonary arterial hypertension (Group 1), pulmonary hypertension associated with left-sided heart failure (Group 2), pulmonary hypertension secondary to lung disease (Group 3) and multifactorial pulmonary hypertension (Group 5). The treatment for CTEPH includes pulmonary endarterectomy ((PEA), class—I, evidence level—B) and/or balloon pulmonary angioplasty ((BPA), class—I, evidence level—B), complimented with drug therapy, if required [8,9]. It is essential to accurately note, label and document the stenotic and occluded lobar and segmental arteries during the diagnostic procedure to ensure the best possible treatment. In treatment via BPA, identical imaging modalities are employed; therefore, explicit preparation, which is essential during diagnostic PA, is crucial. Although the rate of complications in BPA has decreased in past years, it remains high.

The rate of complications related to pulmonary injury varies from 5.9% to 31.4% [10,11], while the early mortality rate is considered to be 0–14%, depending on the experience of the respective CTEPH clinical center [12]. Serious but quite common adverse events include catheter wire-caused vascular injuries, leading to hemoptysis, vascular dissection and balloon-induced perforation of pulmonary arteries [13]. Older age and higher mean pulmonary artery pressure are often associated with a higher risk of complications [14,15]. Unexpected punctures or perforations might be related to the inability to correctly locate the course of a segmental artery, misunderstanding the bifurcation angles in different planes, or misinterpreting the original diameter of the artery as it courses dorsally.

It is essential to emphasize the significance of anatomical variations and arterial morphology in these complications. In addition, a better understanding of topographical arterial anatomy would allow the invasive specialist to perform the PA and BPA procedures quicker, decreasing the risk of complications and the dose of radiation absorbed.

This study aimed to obtain and present a detailed insight into the anatomy of pulmonary arteries for clinical specialists. In clinical environment, pulmonary arteries are observed as black and white structures in different scans while looking for pathologies. The anatomy of pulmonary arteries is of the utmost importance in thoracic surgery, yet it usually concerns only a few pulmonary segments during the surgery. We focused on the pulmonary arteries’ morphological variations, observing both whole lungs in a cadaveric sample. It will also be a great tool for internal medicine specialists to extend their knowledge of pulmonology and improve their understanding of the manifestations of different diseases.

## 2. Materials and Methods

### 2.1. Study Design

From October 2023 to May 2024, an anatomical assessment of the pulmonary arteries was carried out using a pair of human embalmed cadaver lungs provided by the Laboratory of Anatomy and the Department of Morphology of the Institute of Anatomy and Anthropology of Rīga Stradiņš University. The lung specimen was acquired from a male who died of a condition unrelated to lung disease at the age of 93. Outside of the lung, the tissue was covered with some minor gray spots, which could be associated with a lifetime spent in a city and/or minor smoking habits. The normal diameter of the pulmonary trunk (29.70 mm) is also a good indicator of the absence of prominent pulmonary hypertension during the cadaver’s lifetime. No morphological evidence of cardiovascular disease was found in the dissected case; however, a more detailed analysis of the heart would be necessary to rule out all the cardiac pathologies. The cadaver was found to have hepatosplenomegaly. The cadaver’s use of medications that could influence the functions of lungs and pulmonary arteries during its lifetime is unknown. A relative’s consent was obtained before the cadaver was used for scientific purposes. This study involved analyzing the branching patterns of the pulmonary arteries and macroscopically observing, studying and photographing the anatomical distributions. The methods and procedures complied with the applicable instructions, guidelines and regulations. Approval from the Research Ethics Committee of Rīga Stradiņš University was received (ethical approval code: 2-PĒK-4/153/2024).

### 2.2. Dissection Procedure

The thoracic cavity of the male cadaver, which was fixed with the classical formaldehyde method, was opened first. For this part of the dissection, a method described by Barberini was used [16]. In this region, the skin of the thorax was removed by scalpel, and then, sternotomy was performed. A bone saw was used to make a vertical incision through the anterior aspect of the ribs, which were folded back and to both sides, revealing the lungs with the heart between them. The pericardium was peeled off, and the lungs were separated from the diaphragm using a scalpel and forceps. To continue the dissection, the lung–heart complex needed to be extracted from the body. The superior and inferior caval veins were cut off at the height of the right atrium, and the ascending aorta was cut off right above the base of the outflow tract. After removing the lungs from the body, the pulmonary trunk was cut off at its base. The heart was separated entirely from the lungs. The bifurcation of the pulmonary trunk was easily visualized.

Further dissection included removing lung parenchyma, bronchi, veins and nerves and preserving only arteries. The principal bronchi and pulmonary veins were cut off straight away to improve the accessibility for the dissection.

The dissection procedure was based on the principle of following the respective pulmonary artery and removing the lung tissue millimeter by millimeter so as not to cut off some arterial branches accidentally. The process started at the hilum of the lung and continued ventrally, laterally and dorsally in the lung periphery. At the segmental level, the bronchi were accompanying arteries, which made it easier to dissect; removing these bronchi was sometimes effortless, revealing a clear path for the partner artery. In the middle of the dissection, for better accessibility at the bifurcation of the pulmonary trunk, the right pulmonary artery was cut off, which resulted in right and left lung separation, and the dissection was continued for the two separate parts. The dissection was stopped after revealing all the segmental and most of the subsegmental arteries, in consideration of the fragility of the lungs and arteries. Red acrylic paint and a small brush were used to paint all the arteries down to the subsegmental level to emphasize the anatomy of the pulmonary arteries.

### 2.3. Evaluation of the Arterial Branching, Measurements and Comparisons

After the dissection, the branching variance in both lungs was assessed. The non-arterial tissue were removed using forceps and a scalpel to expose the pulmonary arterial branches fully. The recognition of pulmonary segmental arteries is indicated by the letter A, representing the artery, followed by a number corresponding to the pulmonary segments they serve. In cases where two separate arteries supply the same pulmonary segment and do not have a common origin, a lowercase letter (either a or b) is used after the number to denote the subsegmental branch. For example, the anterior segment (S3) is supplied by A3a and A3b.

The identified arterial branches were then measured accordingly. The morphological parameters (length, diameter of each artery and also the angles of bifurcation between the arteries) were detected by a plastic ruler, digital caliper and protractor. All these measurements were taken by hand, each distance and angle was measured five times, and the mean value was taken as the final measurement.

Data from the scientific literature (selective search of articles from PubMed and Scopus (published from 1956 to 2024)) were used and analyzed to compare pulmonary artery branching patterns. In the search for articles, the following keywords were used: “pulmonary arterial tree”, “branching of pulmonary arteries”, “morphology and morphometry of pulmonary arteries” and “branching pattern/variations in pulmonary arteries”. The articles were selected based on their titles and abstracts, and complete text articles were carefully evaluated with the language requirement of being published in English. Some articles were excluded because the pulmonary arteries were described at a more distal subsegmental level, including only one lung lobe. The included articles describe the branching of lobar and segmental arteries. For this research, all articles containing detailed descriptions of the morphometry of pulmonary arteries up to the subsegmental branches were included.

Related to segmental arteries’ variations, Michaud’s approach is rooted in the concept of its description [17], where each lung is divided into ten segmental zones. In the right lung, the bronchopulmonary segments consist of several subunits, including the apical (A1), posterior (A2), anterior (A3), lateral (A4), medial (A5), superior (A6), medial basal (A7), anterior basal (A8), lateral basal (A9) and posterior basal (A10) segments in the right lung. In the left lung, the bronchopulmonary segments include the apicoposterior (A1 + 2), anterior (A3), superior lingular (A4), inferior lingular (A5), superior (A6), anteromedial basal (A7 + 8), lateral basal (A9) and posterior basal (A10) segments.

### 2.4. Statistical Methods

An evaluation of the morphometry of the pulmonary arteries was carried out using Microsoft Excel (2016) to examine the distribution of the data. For this evaluation, all the pulmonary arteries from both lungs up to the segmental level were put together and grouped by the branching generation they belong to, with the pulmonary trunk being the first-generation artery. The distribution of the data was represented through the mean and standard deviation (SD). The coefficient of variation (CV) was computed to compare measurements of different types and units, serving as a statistical indicator of the relative scattering of data points in a data set around an average. Data regarding the length and width of arteries were tabulated, demonstrating the artery’s generation and morphometric parameters like length and diameter. The mean value presents the mean length and diameter of each generation of branching. SD and CV reveal the tendencies (if any are present) in each branching generation’s morphological parameters.

## 3. Results

This study provides a thorough account of the branching arrangement in a single dissected human sample. Each lung can exhibit as many as ten observable segmental arteries. It also emphasized that the branching configurations of the pulmonary arteries should be regarded as distinct entities on each side and not be compared directly.

Figure 1 presents the result of human cadaver lung dissection for preserving the pulmonary arteries which are marked in red. The pulmonary arteries in the left and right lungs form two distinct patterns.

### 3.1. Branching Variant of the Right Lung

The anatomical structure and division of the right lung are illustrated in Figure 2a,b. The right pulmonary artery (RPA) splits into the anterior trunk and interlobar artery after the RPA courses horizontally for a few centimeters. The anterior trunk is frequently observed in the right lung and is more clearly visible in the AP projection, as Figure 2b depicts. It provides most of the upper lobe’s blood supply, delivering A1, which supplies the apical segment with numerous subsegmental branches: A2a, responsible for the posterior segment, and two branches, A3a and A3b, which provide blood to the anterior segment. In the upper lobe, the arteries are very densely packed and overlap; this is a prime example for the necessity of the three-dimensional (3D) model. The interlobar artery then courses caudally and dorsally. The first branch it gives off is A2b, which turns upwards to the posterior segment. So, the upper lobe’s three segments are supplied with oxygen-poor blood via the anterior trunk and A2b (posterior segmental artery). The interlobar artery gives rise to additional arteries at the level of the middle lobe. Ventrally, it serves as the origin of the common trunk of A4 and A5, which subsequently diverge into A4 and A5, supplying the corresponding segments of the middle lobe. A4 follows a more lateral course, while A5 takes a medial trajectory one. Opposite to this trunk, A6 courses dorsally supply the superior segment. The A4 + 5 trunk branches off at the same level as A6, and the branching angles between the interlobar artery and A4 + 5 and the interlobar artery and A6 are nearly perpendicular. The interlobar artery terminates in three branches: A7, A8 and a trunk of A9 + 10. When looking at the arteries from the front, it can be quite challenging to differentiate between these lower lobe arteries. In such situations, just like when examining the apical part of the lung, a 3D dissected model can be beneficial.

In Figure 2a, the medial view clarifies the arteries’ differentiation. A7 runs towards the viewer, and A8 supplies the anterior part of the lower lobe and bifurcates into A9 and A10. A9 extends laterally, while A10 courses posteriorly and medially. The trifurcation of the interlobar artery occurs immediately after it enters the lower lobe, identifiable in Figure 2a due to the prominent fissure between the middle and lower lobes.

### 3.2. Branching Variant of the Left Lung

In Figure 3a,b, the pulmonary arteries of the left lung are depicted. Unlike the right lung, there is no anterior trunk present. In this dissection, the upper lobe receives blood from five individual arteries: three arteries supply the apicoposterior (S1,2) segment, and two supply the anterior (S3) segment. After branching off to S1,2 and S3, the left pulmonary artery (LPA) transitions into the interlobar artery. The initial branches of the artery are known as A3a, A3b and A1. These branches further split into smaller arteries, known as subsegmental arteries. The interlobar artery extends downward and toward the back, giving two more separate branches, A2a and A2b, which supply blood to the apicoposterior segment. As it continues downward, the interlobar artery branches into S4 and S5. A central branch called A4 + 5 also separates from the interlobar artery towards the front. The common trunk divides into two branches, A4 and A5, with A4 running just above A5. The A6 branches off the interlobar artery and then descends to supply blood to the superior lung segment (S6). The interlobar artery ends in three branches (labeled A8, A9 and A10), which provide blood to S8, S9 and S10, covering the entire basal part of the lung. A8 takes a medial and ventral course and gives rise to a small A7 branch that supplies the reduced or absent medial basal segment (S7). A9 supplies the lateral basal part of the lung and extends posteriorly to A10.

### 3.3. Morphological Parameters of the Dissected Model

This study assessed the morphological parameters of pulmonary arteries, examining the length and lumen diameter from the pulmonary trunk to the segmental arteries. The pulmonary trunk was categorized as the first-generation artery. Twenty segmental arteries were included in the analysis. It is important to note that not all segmental arteries are part of the same generation of branching. This study observed that the segmental arteries were primarily of the third and fourth generations, with eight arteries belonging to each category out of the total 20 identified. Additionally, four segmental arteries were classified as fifth-generation.

Table 1 and Table 2 detail the arterial development stages for lung segments. The diameters and lengths of these arteries vary widely. The narrowest artery, A8 of the left lung, measures 3.04 mm wide, while the broadest artery, A1 of the right lung, measures 9.29 mm. In terms of length, the shortest artery is A3 of the left lung, measuring 9.09 mm, and the longest is A4 of the right lung, measuring 53.91 mm.

The segmental arteries were analyzed based on their generation rather than being grouped together, which provides a more precise anatomical view. This approach accounts for the size differences among the arteries and offers valuable insights for anatomical studies.

The mean length values for each branching generation of the dissected case are presented in Table 3. In the model, it was observed that contrary to expectations, the third-generation arteries were, on average, the shortest. These arteries, which branch off the right and left pulmonary or interlobar arteries, typically act as trunks with a relatively small length-to-width (diameter) ratio. For example, A3b is a third-generation artery that appears short but gives rise to long subsegmental branches. Similarly, the common trunk of A4 + A5, a third-generation artery, bifurcates into longer A4 and A5 fourth-generation arteries.

Table 4 shows the relationship between artery generation and mean diameter. There is a substantial decrease in diameter from the second-generation (right and left pulmonary arteries and interlobar arteries) to the third-generation arteries, with the subsequent reduction being more gradual. Figure 1 demonstrates how arteries progressively shrink with each bifurcation as they extend distally within the lung.

Table 3 and Table 4 present the morphometric analysis of pulmonary arteries, which involves grouping segmental arteries by generation for the most precise comparison. A detailed distribution of which segmental arteries belong to which branching generation is presented in Table 1 and Table 2. Additionally, in Table 3 and Table 4, the measurements included are the lobar arteries and common trunks of segmental arteries that belong to second to fifth generation and are not shown in Table 1 and Table 2. Measurements in Table 3 and Table 4 represent the whole pulmonary arterial tree in both lungs up until the fifth-generation arteries. The mean values, standard deviations (SD) and coefficient of variation (CV) are presented.

In Table 3, the coefficient of variation exceeding 30% indicates significant dispersion in artery lengths within the same generation, potentially misleading mean values. This means that there is no clear mean value for the proximal part of pulmonary vasculature; arteries of same branching order have different lengths in different pulmonary segments. However, Table 4 displays a neater pattern with a coefficient of variation below 30% for all generations (2nd to 5th), providing a more reliable insight into changes in arterial diameters. Each branching generation in both lungs represents a similar reduction in the diameter of the following artery.

Three types of branching angles were measured and compared in a recent study. The first measurements included angles between the interlobar artery and its lateral branches, such as separate segmental arteries and standard trunks (e.g., the trunk of A4 + A5). The second set of measurements focused on angles formed by different segmental arteries originating from a single bifurcation, including angles between A4 and A5, A8 and A9, A9 and A10, and the arteries of the upper lobe. Finally, the third set of measurements involved angles formed by bifurcations of the segmental arteries, specifically for all successfully dissected subsegmental arteries. Mean values for each group of angles are presented in Table 5, showcasing the comparative analysis of these branching angles.

The proximal branching angles of the interlobar artery are wide and close to perpendicular, indicating the presence of redundant branches, which are prominent in interlobar artery branching. Dichotomous branching is absent at this point. Moving more distally, the angles between the segmental arteries are closer to 45 degrees, indicating dichotomous branching, with these arteries being similar in size. The most distal angles between the subsegmental arteries are even narrower, close to 30 degrees and exhibit typical dichotomous branching.

## 4. Discussion

Pulmonary vasculature anatomy varies among patients, including differences in the size and angulation of vessels. It is crucial to accurately differentiate between the pulmonary arteries and veins (A/V) when diagnosing and treating pulmonary conditions [18]. A thorough physical examination can detect around 75% of cases in high-risk populations [19]. Fibrotic transformation of pulmonary artery thromboembolism results in chronic obstruction in the macroscopic pulmonary arteries and vascular remodeling in the microvasculature of the pulmonary system [20]. Pulmonary artery pressure monitors enable remote assessment of cardiopulmonary hemodynamics, allowing for early intervention, and this has been proven to reduce hospitalization due to heart failure [21,22].

Increased peripheral pulmonary vascular resistance can significantly impact cases of pulmonary hypertension (PH) [23]. The right main pulmonary artery runs horizontally in the human body and is presented perpendicular to the right lung. This position creates a right angle with the pulmonary artery trunk, potentially leading to smoother blood flow and increased vulnerability to the effects of PH. On the other hand, the curved trajectory of the left main pulmonary artery may make it comparatively less susceptible to the impact of PH when compared to the right main pulmonary artery. In a study with animals, Harper et al. [24] state that the pulmonary artery is not a source of increased resistance.

A wide range of congenital and acquired conditions can affect the pulmonary arteries, some of which are familiar in clinical practice and others rare. Organized blood clots cause chronic thromboembolic pulmonary hypertension (CTEPH) in the pulmonary arteries. Diagnosis can take 1 to 2 years, which results in higher mortality rates [25,26]. It is recommended that every patient undergoes evaluation at a specialized center with experience in pulmonary arterial endarterectomy (PEA), a potentially curative surgical technique [27]. Patients that are considered inoperable (up to 1/3 of patients) or have blockages in distal parts of the pulmonary arterial tree (segmental and subsegmental arteries) undergo a different type of treatment, balloon pulmonary angioplasty (BPA), which is performed in 2D radiographic control [28]. In BPA, knowledge of the topographical anatomy of pulmonary arteries is key to successful treatment via the dilation of narrowed arteries. CTEPH is determined when there is a sustained increase in mean pulmonary artery pressure (at least 20 mm Hg at rest) and evidence of chronic pulmonary embolism on CT, MRI, V/Q scan or PA. Multidetector computed tomography (CT) angiography is typically used to identify indirect (an uneven blood flow pattern in the lungs and the presence of enlarged bronchial arteries) and direct (organized blood clots, partially filled or wholly blocked pulmonary arteries, and thin bands and membranes) signs of CTEPH [9]. Mahammedi et al. [29] found a significant relationship between CT scan measurements of the pulmonary arteries and the severity of PH. MRI is a sensitive and reliable tool for assessing the diagnosis of CTEPH; however, it is rarely used in practice due to the high cost of the examination. V/Q scan is usually performed to rule out CTEPH and diagnose other forms of pulmonary hypertension. It does not reveal specific lobar and segmental arterial branching and precise occlusions, yet it clearly demonstrates, with high sensitivity, perfusion defects in lungs that are present in cases of CTEPH.

Diagnosis of CTEPH typically requires invasive pulmonary angiography and right heart catheterization to assess pulmonary arterial pressure [30]. A chronic thromboembolic disease (CTED) diagnosis, distinct from CTEPH, is made if clotting is present without elevated mean pulmonary arterial pressure. As the pulmonary arterial pressures are not determined in CT pulmonary angiography (CTPA), it is not possible to differentiate CTED from CTEPH via CTPA. In addition, it is crucial to have a highly experienced radiologist review the CT angiography results, as the test is known to have low sensitivity [31].

The field of PH would greatly benefit from more well-designed studies to improve our understanding of how various conditions contribute to disease development [32]. Evidence has shown that delays in diagnosing and managing CTEPH are associated with poor outcomes [33]. Conventional two-dimensional (2D) imaging–pulmonary angiography may introduce several potential errors when assessing the pulmonary vasculature. However, three-dimensional (3D) chest reconstruction, which involves converting 2D imaging data from CT scans into virtual 3D structures using specialized 3D visualization software, provides a more comprehensive and detailed view of the chest anatomy. Radiography can often reveal enlarged central pulmonary arteries and identify congenital pathology with feasible measurements for clinicians without intravascular contrast material or specialized software [34]. It is important to note that although CT pulmonary angiography (CTPA) is a valuable tool, there are more specific and conclusive methods for diagnosing CTEPH. In routine practice, the first step that leads to the diagnosis of CTEPH is a V/Q scan, which is then followed by right heart catheterization and pulmonary angiography to confirm the diagnosis and assess the severity of the disease.

Comprehending the intricate structure and standard imaging features of the pulmonary arterial tree is essential. Pathologies impacting the pulmonary artery can be categorized into five primary groups: congenital conditions, pulmonary artery dilatation, narrowing, filling defects and PH [35]. These conditions are generally identifiable through chest CT or magnetic resonance imaging (MRI). Treatment options exist for almost all variations of this disease [36].

Acquired pulmonary artery variations can lead to different symptoms and expressions. CTEPH often involves the central vasculature, which affects the segmental or subsegmental vessels in other cases. Radiologists and medical specialists should identify vascular differences and communicate their significance to surgeons [37]. During a pulmonary lobectomy, particularly in video-assisted thoracic surgery (VATS), surgeons may encounter technical challenges stemming from variations in the anatomy of the pulmonary arterial tree [38]. Such anatomical diversity can pose complex and intricate obstacles that require careful navigation during the surgical procedure. Performing the intricate dissection, isolation and closure of the artery branches in a lung lobe to be resected is also one of the most complex steps in a standard pulmonary lobectomy.

These posed difficulties emerge from differences in pulmonary segments due to vascular anatomical variations. Typically, there are 10 pulmonary segments in the right lung and 8 in the left. Segmental and subsegmental arteries usually accompany bronchi and are named accordingly, while pulmonary veins run through interlobular septa [39,40]. It must be noted that pulmonary segmental arteries can be more than this exact number in cases when multiple separate arteries supply a single segment. In the left lung, often (in our case as well) arteries are marked similarly to the right lung with numbers from 1 to 10. It is important to understand that A1 and A2 are different segmental arteries but supply the same (apicoposterior) segment in the left lung. The same goes for A7 and A8: they supply the anteromedial segment as the segment is determined by the connective tissue septa, not by the precise number of arteries it supplies. Accessory segmental bronchi have been found by different authors; however, they end blindly and do not supply an additional segment [41]. If all the pulmonary segments are present in a case, these pulmonary segments are still never identical in different lungs. Chen et al. found that the volume proportions of pulmonary segments in a single lobe differ from one case to another [42]. In the right upper lobe, most commonly (74.6%), the anterior segment (S3) dominates or has the highest volume. The authors also state a positive correlation between pulmonary segments’ volume and vascular supply, which seems reasonable. In our case, S3 was supplied by two segmental arteries (A3a and A3b). That might be a valid indicator that S3 is this study’s dominant segment of the right upper lobe. Unfortunately, it was nearly impossible to locate the fibrous septa that separate the pulmonary segments in the dissected material to confirm this assumption. Recognition of the segments’ proportional differences is crucial in pulmonary segmentectomies.

Anatomical cadaveric dissections have long been integral to research, significantly enhancing the understanding of vascular anatomy for many years. Only a handful of studies in the existing literature delve into lung dissections focused on preserving arterial trees. In contrast, others concentrate on elucidating the morphology of the tracheobronchial tree [43]. The most common branching variations described superficially in most articles give a basic understanding of pulmonary arterial anatomy by describing the segmental branches of each lobe [39]. Murlimanju et al. [44] observed that the variations were higher in the left than in the right lung. Their study detected variations in 16.1% of the right lungs and 48.2% of the left lungs. George et al. [45] found that 67.69% of right lungs have two pulmonary arteries, whereas 3.07% possess three. Other researchers offer more detailed and well-explained branching variations in a single lobe. There are often up to eight different branching variations described in a lobe [42,46,47,48]. Unfortunately, for this research, it was impossible to combine all these variations as in each of the articles, only a single lobe was described, and the criteria for each author in differentiating segmental arteries may be different. Research performed by Michaud et al. [17], which fully describes both lungs, is quite detailed. It presents up to three branching variations for each lobe or cluster of segments. The most common variations usually represent 70–75% of cases, rarer variants were noted in 14–15%, and the rarest occur in less than 10% of cases.

Advanced technology enables the utilization of specialized software to measure digital vascular parameters. Singhal et al. [49] have developed a technique for organizing these measurements using the Strahler order concept. According to this concept, the most distal arteries are designated as first-order. When two arteries of equal size connect, the resulting artery is categorized as a second-order artery. This classification continues for subsequent connections. Our study adjusted the numbering system due to the inaccessibility of the smallest and most distal arteries. For instance, the pulmonary trunk was considered a first-generation artery, with its primary branches labeled as second-generation, and so on. While this approach has been utilized in prior studies, it may not be as comprehensive as the Strahler order method [50,51].

First, it is essential to note the branching pattern of the right and left pulmonary arteries. They form two distinct structures that cannot be compared, and assumptions should not be made in cases of low-quality PA. There are segmental differences between the lungs. As the segments are supplied by segmental pulmonary arteries running primarily parallel to segmental bronchi, the venous blood supply differs between the lungs [43]. The dissected case shows some excellent examples. After coursing horizontally for a few centimeters, the right pulmonary artery splits into the anterior trunk and interlobar artery.

In contrast, the left pulmonary artery does not split into two major branches: the anterior trunk is absent. It is a common finding, and the anterior trunk is not recognized as a structure in the left lung [17,39]. In cases where segmental branches, such as S4 and S5, share the exact numbering, it is essential to understand the topography of pulmonary segments. S4 and S5 have different lung positions. Thereby, the course of segmental arteries will be different in both: In the right lung from the common trunk of A4 + A5, A4 branches off laterally, while A5 branches off medially; however, in the left lung, A4 branches off superiorly, while A5 branches off inferiorly. Another reason for not comparing the two branching patterns is apparent in the dissected case. The interlobar artery terminates in a trifurcation in both lungs, exhibiting a bilaterally symmetrical pattern. However, these three end branches are not the same in both lungs. In the right lung are A7, A8 and the trunk of A9 + 10, but in the left are A8, A9 and A10. The dissected model determined that the revealed subsegmental arteries course to the location of each pulmonary segment. However, this may cause difficulties in reading PAs, especially if they are performed in only one projection. It could lead to an imprecise diagnosis and a higher probability of complications during procedures.

As conventional pulmonary angiography is a 2D picturing examination, many factors can contribute to an inaccurate reading of the angiogram. Often, the arteries overlap, and it might be challenging to differentiate them, especially in the lower lobes of the lungs. In addition, many branching variations are recorded: up to eight in each lobe [17,28,42,46,47,48]. A few rare variations were recorded in the dissected case based on Michaud’s representation. Most commonly, the interlobar artery in both lungs ends in bifurcation: A7 + A8 and A9 + A10 in the right lung and in A8 and A10 + A9 in the left lung. In our case, trifurcations, as mentioned above, were noted. Also, although the anterior trunk is a common finding in the upper lobe of the right lung, it can be absent [17].

The variability of further branching is vast after the pulmonary trunk splits into the right and left pulmonary arteries. There are no two identical lungs, as morphological variations include not only different patterns of branching but also different angulation of the same branching patterns. Various authors have noted a high variability of branching patterns in the upper lobes of the lungs [39]. Also, some researchers acknowledge that more variations are found in the left lung [52]. The right upper lobe has three common patterns: (1) anterior trunk + A2; (2) trifurcation of the anterior trunk that supplies the whole lobe; and (3) three separate branches to three respective segments. There can even be accessory recurrent branches from the interlobar artery. A4 and A5, which supply the middle lobe, can branch off the interlobar artery separately, but more commonly, a common trunk of A4 + A5 branches off the interlobar artery. Rarely are there any other arteries supplying the middle lobe; however, one must be careful with recurrent branches that arise from the region of middle lobe and supply the upper lobe. Multiple variations are noted in the right lower lobe as well. One branch often supplies the superior segment (S6); however, there can be two separate branches in some rarer cases. The basal part of the lung is supplied by four arteries: A7 to A10 according to respective segments. These four branches are end branches of the interlobar artery, and the interlobar artery can split very differently. Usually, it is divided into two trunks, each split into two segmental arteries. Many branching variations here include different combinations of trifurcations. To supply the four basal segments, one of the arteries, after trifurcation, bifurcates again. Clinically, it is often hard to differentiate which branch supplies which segment; thus, imaging from at least two projections is required in pulmonary angiography. It may help that in 70% of cases, the first branch that comes off the interlobar artery is A7. Kandathil et al. note that the left upper lobe can be supplied by 2 to 7 arteries [39]. This fact is the reason for many variations in the left upper lobe. Arteries supplying the apicoposterior and anterior segments can come from a common trunk or segmental branches branch off the left pulmonary artery separately, and often, more than one branch supplies the single segment. Like in the right lung, mostly A4 and A5 come off a common trunk of interlobar artery (Kandathil et al. state this possibility as 80%), but they can be separate branches as well. The arteries of the left lower lobe are similar to the right lower lobe: a single A6 is present, sometimes two branches supply S6. In the most common case, the interlobar artery bifurcates and supplies the anteromedial, lateral and posterior segments. Once again, trifurcation is a possibility. For the left lung, one must remember that there are three basal segments, not four, although sometimes a separate A7 can be noted that comes off A8 medially; our case is a great example of that [17,39]. There is some very thorough research on analysis of separate lobes’ variations in subsegmental arteries’ morphology [46,47,48]; however, in the diagnosis of CTEPH and surgical intervention (pulmonary endarterectomy) and balloon pulmonary angioplasty, an understanding of lobar and segmental levels is paramount as more distal arteries are much harder to access. The benefit of mechanically treating them is lower. Usually, there are four pulmonary veins from which blood returns to the heart, flowing into the left atrium. Yet, variable morphological patterns are also possible here, like additional venous branches connecting pulmonary vasculature with the left atrium, one common pulmonary vein from the left lung and others [39].

Furthermore, for patients with severe CTEPH and thromboembolic blockages, low cardiac output and inability to hold their breath for a few seconds, the quality of pulmonary angiogram can be deficient. In such situations, the invasive specialist must be aware of the possible anatomical variations and differences in the course of the arteries in these variations. Three-dimensional models could be a helpful tool for studying the possible variations, differentiating the segmental arteries and looking at different artery courses in other models. Acquiring data from CT pulmonary angiography is another option for a comprehensive study of 3D anatomy. However, a tangible dissected case represents the most accurate variant of human pulmonary arteries with a size ratio 1:1.

The importance of morphological parameters in diagnosis, pathophysiology, and treatment is not described much in the literature. There is some proof of dysregulation in endothelial cells when shear stress on the vessel wall in pulmonary arteries is present [53]. Shear stress can also be caused by unusual branching patterns in the pulmonary vasculature. It is not yet known what angles and in which lobes of the lungs would be predisposed to CTEPH due to shear stress (or maybe other factors). However, if systematic evaluations of each pulmonary angiography were to be performed, some conclusions could be drawn; this is the aim for future studies. Abbasi et al. [54] discovered that there is a significant decrease in the primary pulmonary artery bifurcation angle in patients developing CTEPH.

The branching angles have their role in complications in diagnosing, treating CTEPH and right heart catheterization when the catheter is directed through the pulmonary arteries. Complications include hemoptysis, pulmonary artery perforation and pulmonary artery injury [10,13,15,55]. While specific causes for the injuries are not extensively outlined, unforeseen angling of the pulmonary arteries could be one of the contributing factors. Our results show that some branches of the interlobar arteries form angles close to 90 degrees. As the anatomy course changes direction to almost perpendicular, there is an increased risk of vascular injury when the catheter is inserted into the vessel wall. No evidence has been established regarding the relationship between the angle and complications. However, the invasive specialist should always exercise caution when directing the catheter into the pulmonary arteries. Further research is needed to investigate the complication rate and variations in pulmonary arterial anatomy.

The invasive specialist performing PA and balloon pulmonary angioplasty (BPA) must also be aware of differences in the sizes of pulmonary segmental arteries. Our study showed a vast difference in the diameters of the segmental arteries (3.04–9.29 mm). Other authors have noted quite an extensive variety in segmental arteries’ diameters [49,51]. During BPA, the narrowed segmental artery is expanded by inflating a balloon. The specialist performing the procedure must be conscientious as the segmental arteries, especially having fibrotic/thromboembolic material in their wall, are very fragile and can perforate [15]. As the risk of perforation is high, each specialist must know the differences in morphology of segmental arteries and treat each stenotic or occluded artery separately by not overestimating its original diameter. Improved knowledge and studying of pulmonary arterial morphology might enhance the quality of the provided treatment and decrease the rate of complications.

Summing up the knowledge from the dissected material and other authors’ work in describing pulmonary arterial tree anatomy [17,39,42,46,47,48,56,57], a detailed sketch of the pulmonary arterial tree was meticulously rendered, showcasing intricate and accurate details. The sketch is now used in clinical practice to document precise CTEPH diagnosis by labeling the stenotic or occluded arteries. After the initial PA assessment, the invasive cardiologist identifies and marks the affected arteries. A sketch of the marked arteries is placed in the patient’s medical records for reference in devising further treatment and evaluating the patient’s case. That could be a great practice in every hospital for convenient documentation of the disease’s diagnosis and/or progression. Furthermore, this approach allows for a systematic review of the CTEPH phenotype, identifying the most affected and less commonly affected arteries. That would allow us to do further research on the pathophysiology and risk factors of CTEPH and review already pronounced hypotheses, e.g., about thromboembolic material forming more prominently in the arteries of the right lung [58].

This study also has certain limitations that should be considered. One obstacle is that the natural variations in human anatomy can lead to unpredictability in the results. Tracing pulmonary artery branching at the macroscopic level through intrapulmonary tissue may have limited our understanding of microscopic branching patterns. Another limitation is that we only had access to a single sample of two lungs, preventing us from conducting a comparative analysis. The dissection procedure was also time-consuming. Furthermore, using preserved, embalmed tissues may not accurately replicate the conditions of live situations. The manual revision of the vascular system and the delicate measuring processes may also risk damaging smaller vessels if not performed with maximal care. There are numerous branching variations, so reviewing a single case does not explain all possible variants. The various morphologies of pulmonary arteries are generally not linked to the pathophysiology of several pulmonary diseases, including CTEPH. Consequently, limited research has been dedicated to this area, resulting in a scarcity of detailed information on pulmonary arterial anatomical morphology. The lack of literature on dissected cadaveric pulmonary arteries and their branching patterns further limits the ability to compare our results with those of other studies. Developing an effective system for processing and interpreting the data becomes a paramount concern in light of these complexities, demanding a detailed and comprehensive approach. More extensive research is needed to understand the variations in the population fully.

## 5. Conclusions

Study findings show that interventions in the pulmonary arteries can be challenging due to the high degree of anatomical variability and segmental arteries varying in number, size and origin location. Despite some segments’ similarities and identical names, the branching pattern of the pulmonary arteries differed between the two observed lungs. Identifying the pulmonary segmental arteries for diagnostic purposes can be significantly improved by having topographic recognition of the branching. Knowing which branches supply which segments in each lung is essential, especially in low-quality imaging or in cases of overlapping of the arteries. The dissected model is also a great learning tool for all the clinicians, as no such materials are available and present in the clinical environment. The 3D model, when stated next to 2D pulmonary angiographies acquired during diagnosis of CTEPH, allows interpretation of them to be much more accessible, improving the diagnosis’s accuracy and the treatment’s effectiveness and safety.

## Figures and Tables

**Figure 1 diagnostics-14-01684-f001:**
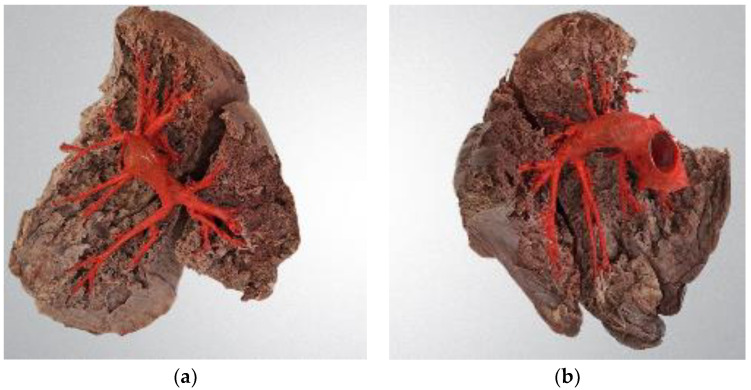
Dissected human lungs: (**a**) right (on the left); (**b**) left (on the right). Pulmonary arteries are colored in red; picture is taken in medial projection.

**Figure 2 diagnostics-14-01684-f002:**
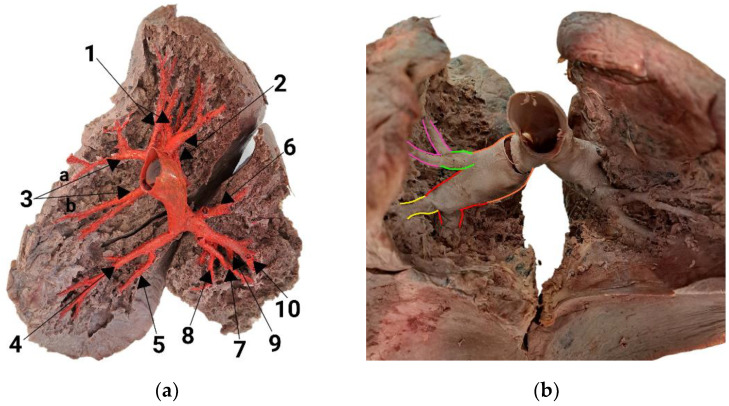
Pulmonary arteries of the right lung: (**a**) medial view; (**b**) anterior view. The numbers adjacent to the segmental arteries denote the specific lung segment supplied by each artery: 1—A1/apical, 2—A2/posterior, 3—A3/anterior, 4—A4/lateral, 5—A5/medial, 6—A6/superior, 7—A7/medial basal, 8—A8/anterior basal, 9—A9/lateral basal, 10—A10/posterior basal. Additionally, different colors represent the arteries: orange—right pulmonary artery; red—interlobar artery; green—anterior trunk; pink—A3/anterior segmental artery; yellow—A4 + A5 trunk/trunk of lateral and medial segmental arteries.

**Figure 3 diagnostics-14-01684-f003:**
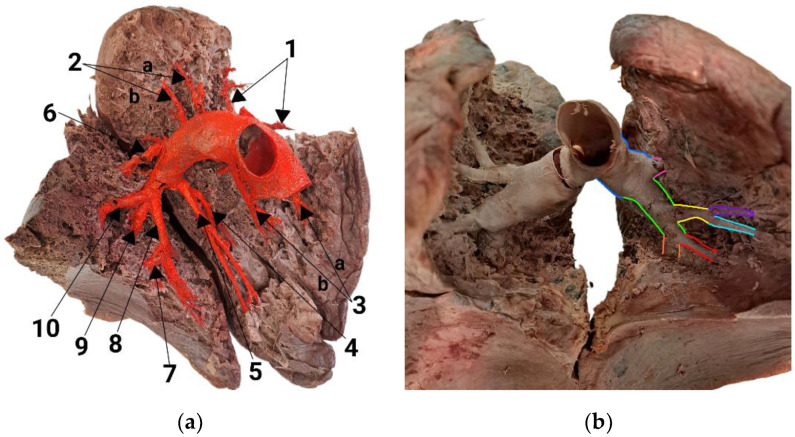
Pulmonary arteries of the left lung: (**a**) medial view; (**b**) anterior view. The numbers following the segmental arteries represent the specific segments that each artery supplies, as follows: 1 and 2—A1,2/apicoposterior, 3—A3/anterior, 4—A4/superior lingular, 5—A5/inferior lingular, 6—A6/superior, 7—A7/medial basal, 8—A8/anterior basal, 9—A9/lateral basal, 10—A10/posterior basal. A2a, A2b, A3a and A3b are pairs of segmental arteries that supply the posterior part of the apicoposterior segment (S1,2) and the anterior segment (S3). The colors representing arteries are as follows: dark blue—left pulmonary artery; green—interlobar artery; pink—A3/anterior segmental artery; yellow—A4 + A5 trunk (trunk of superior and inferior lingular arteries); red—A8/anterior basal segmental artery; orange—A10/posterior basal segmental artery; light blue—A5/inferior lingular artery; purple—A4/superior lingular artery.

**Table 1 diagnostics-14-01684-t001:** The generation, length and diameter of pulmonary arteries in the right lung.

Artery	Generation	Length, mm	Diameter, mm
Pulmonary trunk	1	48.51	29.70
Right pulmonary artery	2	33.15	25.95
A1, apical	3	18.20	9.29
A2, posterior	3	- ^2^	5.87
A3, anterior ^1^	3	29.82	7.81
A4, lateral	4	53.91	5.63
A5, medial	4	36.84	4.84
A6, superior	3	17.87	7.43
A7, medial basal	4	15.93	4.75
A8, anterior basal	4	16.81	7.20
A9, lateral basal	5	37.68	4.34
A10, posterior basal	5	20.46	6.92

^1^ A3 measurements were obtained by averaging the values of A3a and A3b. ^2^ The length of the A2 artery was indeterminable due to its severance during dissection.

**Table 2 diagnostics-14-01684-t002:** The generation, length and diameter of pulmonary arteries in the left lung.

Artery	Generation	Length, mm	Diameter, mm
Pulmonary trunk	1	48.51	29.70
Left pulmonary artery	2	29.46	25.87
A1, apical	3	15.98	4.48
A2, posterior ^1^	3	14.77	3.64
A3, anterior ^1^	3	9.09	7.48
A4, superior lingular	4	19.34	4.79
A5, inferior lingular	4	15.87	4.76
A6, superior	3	19.09	6.65
A7, medial basal	5	12.02	5.25
A8, anterior basal	5	15.78	3.04
A9, lateral basal	4	11.26	6.33
A10, posterior basal	4	28.94	6.42

^1^ Measurements of A2 and A3 were obtained by calculating the mean values of A2a, A2b, A3a and A3b.

**Table 3 diagnostics-14-01684-t003:** Statistical analysis of arteries’ generations and mean lengths.

Generation	Mean, mm	SD, mm	CV, %
2	31.31	1.85	5.89
3	17.83	5.79	32.47
4	24.86	13.44	54.06
5	21.46	9.82	45.69

**Table 4 diagnostics-14-01684-t004:** Statistical analysis of arteries’ generations and mean diameters.

Generation	Mean, mm	SD, mm	CV, %
2	25.91	0.04	0.15
3	6.58	1.73	26.29
4	5.59	0.90	16.03
5	4.89	1.41	28.88

**Table 5 diagnostics-14-01684-t005:** The branching angles of the pulmonary arteries in the right and left lungs.

Branching Angle Group	Mean Angle, Right Lung	Mean Angle, Left Lung
Off the interlobar artery	85.0°	77.0°
Between the segmental arteries	57.5°	38.0°
Between the subsegmental arteries	28.0°	33.6°

## Data Availability

The original contributions presented in the study are included in the article, further inquiries can be directed to the corresponding author.

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
