# Peer review of "The Importance of Topographical Recognition of Pulmonary Arteries in Diagnostics and Treatment of CTEPH, Based on an Analysis of a Dissected Case Model—A Pilot Study"

_diagnostics, 2024, doi:10.3390/diagnostics14151684_

Round 1

Reviewer 1 Report

Comments and Suggestions for Authors

From my point of view, the study is very important for the management of venous thromboembolism and its consequences, such as CTEPH. Therefore, I sincerely appreciate the effort of the Authors to perform such research.

Content suggestions:

1.         I supposed that for the variations of the anatomy of the circulation of the lungs, more samples are needed.

2.         I would like to kindly ask the Authors about the past and drug history of the patient, as this can modify the results.

Reviewer 2 Report

Comments and Suggestions for Authors

Thank you for the possibility to review the manuscript titled: “Importance of Topographical Recognition of Pulmonary Arteries in Diagnostics and Treatment of CTEPH, Based on Analysis of a Dissected Case Model – Pilot Study”. The anatomical evaluation of the lungs is essential for several disciplines such as pulmonology, thoracic surgery, radiology etc. Therefore, this effort is appealing, as dissection studies are a “dying” field. However, there are multiple recommendations to consider:

-The sentence “93-year-old male with a relatively healthy life status” is misleading. A 93-year-old male most likely has arterial problems which involve the lungs as well. Therefore, I would recommend to state that “lung specimen was acquired from a male who died of a condition unrelated to lung disease”. This will indicate that the anatomical specimen is “unchanged” as maximum as it is possible.

-The aim of the study is to apply knowledge from a single case of dissection. This is hard to understand as a single pair of lungs cannot adequately contribute to the overall knowledge of lung anatomy. I think as a pilot study the current aim is to demonstrate the anatomical particularities for clinicians who do not contact with anatomical specimens on a daily basis.

-The statistical evaluation requires specification, particlalry it is hard to understand what was compared as standard deviation and coefficient of variation of different types and units. Did the author compare the measurements between lungs or different segments?

-Lung segments require a separate evaluation in the discussion section as they often depend of the vascular supply.

Please take into account the recommendations in the spirit of improving the quality of the submission.

Round 2

Reviewer 2 Report

Comments and Suggestions for Authors

The authors have made all of the necessary corrections